# Unraveling the Rewired Metabolism in Lung Cancer Using Quantitative NMR Metabolomics

**DOI:** 10.3390/ijms23105602

**Published:** 2022-05-17

**Authors:** Karolien Vanhove, Elien Derveaux, Liesbet Mesotten, Michiel Thomeer, Maarten Criel, Hanne Mariën, Peter Adriaensens

**Affiliations:** 1Applied and Analytical Chemistry, Institute for Materials Research, Hasselt University, Agoralaan 1-Building D, B-3590 Diepenbeek, Belgium; peter.adriaensens@uhasselt.be; 2Department of Respiratory Medicine, AZ Vesalius, Hazelereik 51, B-3700 Tongeren, Belgium; 3Faculty of Medicine and Life Sciences, Hasselt University, Martelarenlaan 42, B-3500 Hasselt, Belgium; elien.derveaux@uhasselt.be (E.D.); hanne.marien@uhasselt.be (H.M.); 4Department of Nuclear Medicine, Ziekenhuis Oost-Limburg, Schiepse Bos 6, B-3600 Genk, Belgium; liesbet.mesotten@zol.be; 5Department of Respiratory Medicine, Ziekenhuis Oost-Limburg, Schiepse Bos 6, B-3600 Genk, Belgium; michiel.thomeer@zol.be (M.T.); maarten.criel@zol.be (M.C.)

**Keywords:** lung cancer, NMR (nuclear magnetic resonance), metabolism

## Abstract

Lung cancer cells are well documented to rewire their metabolism and energy production networks to enable proliferation and survival in a nutrient-poor and hypoxic environment. Although metabolite profiling of blood plasma and tissue is still emerging in omics approaches, several techniques have shown potential in cancer diagnosis. In this paper, the authors describe the alterations in the metabolic phenotype of lung cancer patients. In addition, we focus on the metabolic cooperation between tumor cells and healthy tissue. Furthermore, the authors discuss how metabolomics could improve the management of lung cancer patients.

## 1. Introduction

Lung cancer is the leading cause of cancer mortality worldwide [1]. Globally there were an estimated 2.2 million new lung cancer diagnosis and 1.8 million lung cancer deaths in 2020. The overall survival of patients with lung cancer remains poor as most of the patients are diagnosed at an advanced stage. Several independent randomized controlled clinical trials confirmed the efficacy of annual low-dose computed tomography (LDCT) screening in reducing lung cancer mortality in a high-risk population based on age and smoking history [2]. LDCT-imaging often identifies suspicious lung lesions but cannot verify whether these are the results of benign disease or a truly aggressive malignancy, leading to supplementary imaging techniques and invasive strategies to obtain tissue. Cancer cells have different metabolic programs than normal cells, such as an increased consumption of glucose and glutamine, and increased aerobic glycolysis and lipid metabolism [3,4,5,6]. Nowadays, researchers focus increasingly on the metabolic alterations of cancer cells, and metabolomics has become a powerful emerging technology to study the biochemistry of cancer. Metabolomics is defined as the analysis of small molecule metabolites in tissue, plasma, and other body fluids. Currently, mass spectrometry and ^1^H-NMR (proton nuclear magnetic resonance) are the major tools to analyze a large number of metabolites simultaneously. Metabolic alterations in cancer cells result in different metabolite concentration levels in plasma of patients with lung cancer when compared to controls and patients with inflammatory diseases [7,8,9,10]. Therefore, quantitative NMR metabolic profiling of plasma has the potential to become a minimally invasive tool to explore the need for more invasive procedures in patients with a positive suspicious screening result. Lung cancer survival and treatment are mostly determined by the tumor, nodus, and metastasis (TNM) classification [11]. However, the TNM system is based on anatomic findings and does not reflect the biochemical profile of lung cancer. As a result, the TNM classification does not provide a satisfactory explanation for the differences of survival and therapy response in apparently similarly staged patients. With our growing understanding of cancer biology and advances in molecular technologies, a variety of genetic and molecular features have been proposed as prognostic biomarkers, although the prognostic role of metabolites has not yet been further explored in lung cancer. Prognostic models integrating metabolic data may result in more individualized survival estimates. In addition, predictive models may result in more efficient personalized and targeted therapeutic approaches. In this review, the authors discuss the rewired metabolic pathways in cancer and the principles of quantitative NMR metabolomics. Furthermore, the potential in diagnosis, prognosis, and prediction of treatment response are highlighted. Finally, the authors summarize the targeted treatment of lung cancer based on metabolic vulnerabilities.

## 2. Rewired Metabolic Pathways in Cancer

Metabolism is responsible for deriving energy and biomolecules from the cellular environment in healthy and cancer cells. Metabolic changes, driven by genetic mutations and the tumor microenvironment, are crucial hallmarks of cancer cells [12,13]. In early experiments, Otto Warburg observed that cancer cells secreted most glucose-derived carbon as lactate, even under normal oxygen concentrations [14]. Initially, the Warburg phenomenon was attributed to defective oxidative phosphorylation or mitochondrial biosynthesis, and many investigators assumed that mitochondrial metabolism was a negligible contributor to macromolecule synthesis and energy (ATP, adenosine triphosphate) production. However, more recent data establish that most cancer cells have normally functioning mitochondria capable of oxidative phosphorylation and biosynthesis [15,16]. Multiple studies have indicated increased flux through glycolysis, and the tricarboxylic acid (TCA) cycle generates essential metabolites for macromolecule synthesis [17,18]. In the net yield of ATP, glycolysis is less efficient than oxidative phosphorylation. However, cancer cells adapt to this disadvantage by increasing the uptake of glucose and the upregulation of glycolytic enzymes which facilitates a higher glycolytic flux [19]. In addition, it has been demonstrated that most human cancers also produce ATP through oxidative phosphorylation [20,21]. Apart from the production of ATP that is needed in anabolism, the glycolytic intermediates play a pivotal role in macromolecular biosynthesis. Lung cancer cells upregulate the isoform PKM2 of the enzyme pyruvate kinase (PK), which catalyzes the conversion of phosphoenolpyruvate to pyruvate. This remarkable isoform has a low activity which implies the accumulation of glycolytic intermediates that are subsequently directed into biosynthetic pathways such as the pentose phosphate pathway (PPP), the hexosamine biosynthesis pathway (HBP), the serine biosynthesis, and the one-carbon metabolism [4,22]. The PPP has a pivotal role in promoting cell growth and survival by providing NADPH (nicotinamide adenine dinucleotide phosphate), needed for fatty acid synthesis and detoxification of ROS (reactive oxygen species), and pentose phosphate for nucleotide synthesis. The hexosamine biosynthetic pathway is responsible for the production of the activated monosaccharide UDP-GlcNAc (uridine diphosphate N-acetylglucosamine) This product is used as a substrate in glycosylation reactions of lipids, proteins, or to generate UDP-GlcNAc-derived monosaccharides also used for glycosylation [23]. Aberrant glycosylation, depending on the production of UDP-GlcNAc, is considered a hallmark of cancer, and, as a consequence, it seems reasonable to assume that the HBP is important in tumorigenesis. Complex carbohydrates have impact on cellular signaling and the regulation of cell–cell adhesion and play a role in the cell–matrix interaction. As a consequence, alterations in cellular glycosylation are associated with malignant transformation of cancer cells, tumor progression, and, ultimately, metastasis formation [23,24,25,26]. One-carbon units are derived from glucose and amino acids such as serine and glycine and generate molecules that serve as building blocks for biosynthesis and redox reactions [27]. Furthermore, the one-carbon metabolism has an important role in regulating substrates for epigenetic and post-translational modifications. To fulfill the high anabolic demands of cancer cells, oxidative phosphorylation serves alongside glycolysis, and malignant cells balance between alternative pathways for biosynthesis while maintaining a controlled influx into the TCA cycle. The evidence of both an enhanced glycolysis and TCA cycle was demonstrated in lung cancer patients using a labeled glucose tracer by Hensley et al. [17]. The increased uptake of glucose and its subsequent phosphorylation, resulting from an increase in glucose transporters (GLUT) and upregulation of hexokinase, forms the basis for positron emission tomography (PET) imaging. Indeed, an elevated increase in GLUT transporters is seen in many cancers, including lung cancer [28]. PET-derived parameters, such as the maximum standardized uptake value (SUV_max_), the metabolic tumor volume, and total lesion glycolysis, have been associated with poor outcomes [29]. However, PET-derived parameters do not take alternative fuels, such as amino acids, into account and probably only partly explain cancer’s biological behavior and aggressiveness. Figure 1 summarizes the metabolic alterations in glucose metabolism in lung cancer cells as extensively described by Vanhove et al. [4].

A principal nutrient other than glucose, but required by proliferating cells, is glutamine, the most abundant amino acid in the human blood. As an alternative carbon source, glutamine supplies the growing cell with carbon intermediates to support energy generation and the accumulation of building blocks [30]. Furthermore, as illustrated in Figure 2, glutamine catabolism, i.e., glutaminolysis, provides nitrogen for the biosynthesis of purines, and glutamine-derived carbons synthesize pyrimidines and non-essential amino acids. In addition, glutamine-derived glutamate is a component of the main antioxidant factor glutathione, and glutamine stimulates the expression of uncoupling protein 2 (UCP2) in the mitochondrial membrane and thereby regulates substrate oxidation in the mitochondria [31,32].

In cells without mitochondrial dysfunction, the oxidation of glutamine is the major source of anaplerosis, a phenomenon to replenish intermediates in the Krebs cycle that is redirected into biosynthetic reactions. Hypoxic cancer cells and cells with dysfunctional mitochondria due to TCA cycle enzyme mutations can also provide carbon for fatty acid biosynthesis through the reductive carboxylation of glutamine-derived α-ketoglutarate resulting in the formation of citrate, thereby supporting the biosynthesis of fatty acids [30]. The reductive carboxylation is dependent on cytosolic isocitrate dehydrogenase IDH1 and mitochondrial IDH2, catalyzing the reverse reaction of isocitrate production from α-ketoglutarate. This process not only supports the biosynthesis of macromolecules but also plays a role in redox homeostasis. Jiang et al. even speculated that a disturbance stimulates the reductive carboxylation in the mitochondrial redox ratio [33]. Impairment of the electron transport chain results in a less efficient oxidative phosphorylation as the NAD^+^/NADH (nicotinamide adenine dinucleotide) ratio decreases [33]. Transfer of reducing equivalents from NADH to NADPH by transhydrogenase may drive the NADPH-dependent reductive carboxylation by IDH1 and IDH2. Indeed, the reductive carboxylation by IDH1 results in the production of cytosolic NADP^+^ and isocitrate, while oxidation of mitochondrial isocitrate by IDH2 results in the production of NADPH that in turn supports the oxidative phosphorylation. Indeed, glucose-derived carbon is rather secreted into lactate during hypoxia than fueled into the Krebs cycle. However, the carbon molecules that fuel mitochondrial metabolism in vivo are not fully understood. Faubert et al., using labeled lactate, revealed that lactate fuels the Krebs cycle in human lung tumors [18]. In NSCLC (non-small cell lung cancer), the evidence of lactate utilization was most apparent in cancers with aggressive behavior and high ^18^fluorodeoxyglucose (^18^FDG) uptake with a predominance of the contribution of lactate to the TCA cycle [18]. This evidence suggests that measurements to assess lactate utilization might help predict oncological aggressiveness. In addition, higher expression of the lactate transporters, monocarboxylate transporters MCT1 and MCT4, and the enzyme lactate dehydrogenase (LDH) were documented. Unfortunately, inhibition of LDH or MCT by small molecules has not been successful. Oversaturation of the TCA cycle results in the production of ROS, which are detrimental to the cells [3]. Initially, ROS are advantageous to cancer cells as they initiate DNA (deoxyribonucleic acid) mutations that promote malignant transformation and promote progression and metastasis [34]. To prevent the toxic accumulation of ROS, cancer cells in a hypoxic nutrient-deprived environment increase their antioxidant capacity to maintain the ROS concentration in non-lethal but stimulatory levels [16]. Both glutamine-derived glutamate and glycine (one-carbon metabolism) contribute to redox homeostasis as building blocks of the antioxidant glutathione. Apart from the Warburg effect, there is compelling evidence that cancer cells have alterations in different aspects of lipid metabolism. Most of the acetyl-CoA (acetyl-coenzyme A) used for the de novo synthesis of fatty acids is generated from glucose via converting pyruvate to citrate in the mitochondria. As previously mentioned, glutamine-derived α-ketoglutarate can act as a glucose alternative to produce citrate. Malignant cells are characterized by an increase in lipid production with upregulation of components of fatty acid synthesis [35]. Fatty acids are important in forming lipid bilayers and altering the membrane composition towards an increased percentage of saturated fatty acid that confers resistance to oxidative damage [36]. Furthermore, cholesterol with acetyl-CoA as an essential building block plays an important role in regulating the fluidity and permeability of cell membranes. In addition, cholesterol is essential in lipid raft domains that coordinate the activation of several signal-transduction pathways [37]. Cancer cells may utilize alternative pathways other than transport to acquire nutrients in harsh conditions such as starvation. Indeed, macropinocytosis, i.e., internalization of extracellular components such as, e.g., proteins, has been documented in cancer cells expressing *RAS* genes during nutrient deficiency [38,39]. Last, but not least, continuous communication between malignant and stromal cells creates a complex and dynamic microenvironment [12]. The tumor microenvironment harbors malignant cells and other cell types such as endothelial cells, cancer-associated fibroblasts, and immune cells. These cells can affect cancer cell behavior by providing substrates or competition and thus contributing to the metabolic needs of tumor cells. The complex interaction between the metabolic rewiring of cancer cells and other hallmarks of cancer has been extensively reviewed by Elia et al. [40].

## 3. Quantitative NMR Metabolomics

Although the Warburg effect provides the rationale for PET-CT, an imaging method used to detect cancer cells, a single metabolic alteration is not sufficient to define the rewired metabolism in lung cancer. As mentioned before, cancers also take advantage of alternative substrates such as amino acids, e.g., glutamine and fatty acids, and in addition, cells in the tumor microenvironment (TME) can further affect the cancer cell behavior [40]. Furthermore, the metabolic composition of the plasma becomes even more complex if the scarcity of nutrients results in compensatory mechanisms such as degradation of muscle proteins and adipose tissue to resupply pools of cellular metabolites. The diagnosis, biological behavior, and prognosis of cancer are probably not associated with only one metabolic pathway, and therefore high-throughput metabolic profiling approaches are mandatory. Further elucidation of the metabolic reprogramming may result from large studies that consider genetic changes and the tumor micro environment.

Metabolomics is defined as a study of chemical processes involving small molecule substrates, i.e., metabolites, intermediates, and products of cellular metabolism. The application of metabolomic platforms in the search for cancer biomarkers has increased exponentially over the past decade [7,41,42,43,44,45,46,47]. A frequently used technique to analyze low-molecular-weight molecules in biofluids, such as plasma, is proton nuclear magnetic resonance (^1^H-NMR)-based [48,49]. Each hydrogen nucleus in a different chemical environment has a different, specific resonance frequency and therefore results in a signal at a different position in the NMR spectrum. This signal position is expressed as a chemical shift (in ppm, parts per million) relative to a standard compound such as deuterated trimethylsilyl-2,2,3,3-tetradeuteropropionic acid (TSP). Spectra of blood plasma also display broad lines of proteins and lipids that may overlay with the sharp peaks of small metabolite molecules [50], thereby masking some metabolites’ presence. To overcome this, some groups proposed protocols to remove these macromolecules from the biofluid [51,52,53]. A drawback is that such protocols are time-consuming and generally prone to low reproducibility. Attenuation of the broad signals of macromolecules, without causing deleterious effects on the signals originating from free, low-molecular-mass metabolites, by specific NMR pulse sequences, such as the Carr–Purcell–Meiboom–Gill (CPMG) pulse sequence, results in excellent reproducibility [54].

As demonstrated in Figure 3, NMR spectra are very crowded due to the small chemical shift range over which the metabolite signals of the biofluids appear.

One drawback in identifying metabolites in current practice is the absence of standardized collection and measurement protocols for different biofluids. Varying chemical shift values reported for different matrices or non-human species complicate correct signal assignment [55,56,57]. In addition, the chemical shift values (peak positions) depend on conditions such as pH, concentration, temperature, and ionic strength, resulting in less accurate identification and quantification if not exactly replicated. A way to accurately determine the chemical shifts of the plasma metabolite signals under the experimental conditions used is by spiking plasma samples from a large plasma pool with different, known metabolites [49]. Hereto, a small quantity of a known metabolite is added to a sample aliquot of the plasma pool and this is repeated for all metabolites of interest. If the spiked metabolite is present in the sample, its signal(s) will show an increase in intensity [58]. This way, metabolite spiking enables the identification of all metabolite signals, including those most influenced by the disease. Using a standardized protocol, the NMR platform is a very accurate, quantitative method with high analytical reproducibility. Moreover, NMR requires little sample preparation (only buffer addition), and in contrast with other analytical platforms, such as mass spectrometry, no extra sample preparation steps, such as extraction of hydrophobic/hydrophilic components or chemical derivatizations, are needed. In addition, NMR can be used to identify unknown and unexpected metabolites and is well adapted for measuring large cohorts as a high-throughput method. The high reproducibility and simplicity of sample preparation make NMR applicable in large cohorts of multi-sites and longitudinal studies. Moreover, ^1^H-NMR is suited for high-throughput screening as it takes only a few minutes to acquire a spectrum. However, NMR also has disadvantages, with the most important drawbacks being its intrinsically lower sensitivity and signal overlap. Several approaches to improve the sensitivity, such as higher magnetic field strengths and cryo-cooled probes and microprobes, are available. Improvement in signal overlap can be obtained by using higher magnetic field strength NMR spectrometers. Once acquired, NMR spectra are processed which involves phasing, baseline correction, integration, and normalization. In a next step, the spectrum is divided into multiple integration regions of which the integration values can be used as variables in multivariate statistical models. As described by Derveaux et al., metabolite spiking of biofluid samples results in well-defined integration regions representing a single metabolite, or a combination of several metabolites [49]. The primary goal in metabolomics is to extract discriminating information from complex large datasets.

The outcome of a multivariate analysis provides an overview of the metabolites (variables) that are affected by the disease and thus have the potential to differentiate the study subjects. Pattern recognition and related multivariate statistical techniques, such as shown in Figure 4 for PCA and OPLS-DA, are applied to detect relevant patterns and to identify metabolic signatures having diagnostic or prognostic value [59,60].

Principal component analysis (PCA) is the most widely used method to reduce the dimensional space of the data and provides an overview of the variability in the dataset. PCA summarizes most variation in the dataset into a smaller number of principal components (PC) without a priori knowledge of the sample class, i.e., PCA is an unsupervised approach. Each PC is a weighted linear combination of the original variables, and each consecutive PC describes the maximum additional variation in the data that was not represented by the previous PC. PCA results are reported in score plots and loadings. The score plots provide an overview of all the samples, i.e., each point in the plot corresponds to a sample and enables the visualization of groups, trends, and outliers. Loading plots illustrate which variables have the most important contribution to the positioning of the samples on the scores plot. In this context, they are responsible for the observed clustering of samples as visualized in the score plot. As the directions in both plots correspond, the loadings can explain the clustering of the spectra on the scores plot. In addition, PCA plots permit to identify outliers and can be used to identify clustering patters and dominant variation which may not be associated with the real biological effect but could be associated with a secondary effect such as, e.g., diet, age, gender, and other diseases than aimed for such as diabetes or chronic obstructive pulmonary disease in plots of cancer patients. 

PCA is commonly followed by supervised techniques where class information of samples is used to maximize the separations between different groups of samples and detect the metabolic alteration contributing to this classification. Partial least square (PLS) analysis is a supervised method that links a data matrix of predictors (Y), usually spectral intensity values, to a matrix of responses (X) containing quantitative values. In PLS-discriminant analysis, the response matrix is categorical, i.e., the matrix contains sample class information such as cancer versus healthy. A preprocessing filter is combined with PLS to remove structured noise, i.e., irrelevant parts of the spectra that are not correlated with the response. Orthogonal partial least squares discriminant analysis (OPLS-DA) filters this structured noise resulting from physiological variation (diet, age, gender, etc.) and analytical variation. The horizontal-axis predictive component of the OPLS score scatter plot captures information between the lung cancer and control group. The vertical-axis orthogonal component describes the variation orthogonal to the predictive component and thus captures the variation within the control and within the lung cancer groups. OPLS-DA enhances the interpretation of the model and identifies important values responsible for the observed classification. Loading weights, variable importance on projection, and regression coefficients plots are used to determine the most discriminating variables. Predictability estimates of PLS and OPLS models can be easily overfitted in studies where the number of variables is significant and, consequently, so is the chance of high false correlations. To estimate the ability of the model to predict the Y values of new individuals, the dataset is split into a training and a test set (independent validation set) where the training set is used to build the model and the test set is used to estimate the predictability. However, splitting the dataset results in a model built with only a fraction of the dataset. This implicates that enough participants are needed in these trials. Indeed, the significance of the results depends on sample size. When the number of samples is low or no test set is available, k-fold cross-validation is the primary strategy. During this procedure, a k-subset of samples is iteratively left out and predicted back into the model until all samples have been used once. Different quality parameters such as R2, i.e., the goodness of fit or description of the data by the model, and Q2, i.e., the predictive ability of the model, summarize the results of the procedure. In general, the R2 value increases with the number of components while Q2 reaches a plateau that finally decreases with more components at a certain number. At this point, it is very plausible that the model is trying to fit dataset characteristics that are not representative of the study population. Validation by a permutation test is frequently performed in models with poor Q2 values and results in a p-value that estimates the significance of the model. In addition, a receiver operating characteristic (ROC) curve of the training model is an important evaluation method to check the performance of the classification model. A high area under the curve demonstrates that the model has an excellent capability to distinguish between the different classes. 

Another challenge in omics technologies is the analysis of serial samples measured at different time points or matched samples before or after interventions. Using samples of the same patient cohort to monitor the individual response to an intervention or treatment implicates that these samples are paired and dependent. OPLS-DA results in a trained model and interpretation between two sample sets [61].

Nonetheless, this method does not consider the matched or paired-sample information. Therefore, using OPLS-DA on dependent data might result in less robust models and potentially false negative and false positive discriminatory metabolites [62]. OPLS-effect projections (OPLS-EP) is a novel multivariate statistical analysis strategy that allows paired or dependent analyses of individual effects and can be seen as a multivariate version of the paired t-test. In OPLS-EP, an effect matrix is created by subtracting the pre-treatment data (e.g., pre-intervention; pre-operative) from the posttreatment data (e.g., post-intervention; post-operative) and modeled towards an identical y-response vector for all subjects. As this matrix contains the effect of the intervention on each patient, the variation between (predictive component) and within (orthogonal components) subjects is separated, and intrinsic differences in treatment effect between individuals can be identified. Although the application of ^1^H-NMR to lung cancer research is an emerging field, several research groups have established a metabolic profile for lung cancer [7,8,41,45,49]. However, defining prognostic metabolic profiles remains challenging.

### 3.1. Diagnosis

As demonstrated by several research groups, ^1^H-NMR-based metabolic profiling of plasma can effectively differentiate lung cancer patients from controls [7,8,41]. Additionally, profiling of plasma can effectively differentiate PET-positive benign from malignant lung lesions [9]. Therefore, the metabolic signature might become a promising technique for the non-invasive diagnosis of solitary pulmonary nodules. However, the translation of metabolomics from bench to bedside remains a challenge. Indeed, there are many modifiers of the metabolic phenotype, such as, e.g., gender, age, diseases, and many more. Therefore, aside from a metabolic phenotype of cancer, there is also a need to define the normal plasma metabolic profile as a point of comparison. More specifically, the normal range of inter-and intra-individual metabolites variance needs to be defined. Metabolic profiling has been applied to lung cancer patients and controls to identify a robust classification model with high sensitivity and specificity by our research group. A first model by Louis et al. of ^1^H-NMR-based metabolomics resulted in an OPLS-DA model that allowed correct classification of 78% of the patients with lung cancer and 92% of controls [7].

Moreover, validation in an independent cohort demonstrated a sensitivity of 71%, a specificity of 81%, and an AUC (area under the curve) of 0.84 [7]. Interestingly, this model discriminated between patients with early-stage lung cancer (stage I) and a randomly selected group of controls. Indeed, the ^1^H-NMR-derived metabolic profile indicated that metabolic alterations are present even in the initial phase of cancer development. The metabolites that contributed the most to the groups’ differentiation were glucose, glycerol, lactate, N-acetylated glycoproteins, β-hydroxybutyrate, leucine, lysine, tyrosine, threonine, glutamine, valine, aspartate, alanine, sphingomyelin, citrate, phosphatidylcholine, and other non-cholinated phospholipids. In contrast with the observation of Warburg, the plasma metabolic profile of lung cancer patients was characterized by elevated glucose and decreased lactate levels. This finding is in line with the results of Chen and colleagues [63]. Indeed, the findings in plasma were that homeostasis serves to deliver metabolic precursors just in time and rather mirrors the Warburg effect that originated from findings in cancer cells [64]. Homeostasis is defined as the ability of an organism to maintain a specific steady internal condition, such as concentrations of metabolic precursors, intermediates, and metabolic products. This homeostatic tendency of the human body and variation of metabolite plasma concentrations by other confounders, such as age, gender, genetic background, health status, diet, activity levels, and diurnal variations, are even more challenging in the development of a specific lung cancer metabolic profile with diagnostic and prognostic biomarkers [46]. More studies on the contributions to the confounders, as mentioned earlier, are urgently needed and will require significant technical, financial, and human resources.

The increased plasma levels of glucose and decreased levels of both lactate and alanine in malignant cells suggest increased gluconeogenesis as both alanine and lactate are essential gluconeogenic precursors. Initially, lactate was considered a waste product derived from anaerobic glycolysis. Recent work of Hui et al. and Faubert et al. revealed that lactate is a primary fuel in the Krebs cycle. Hui et al. demonstrated that circulating lactate contributed more than glucose in mice models of lung cancer [65]. Using labeled nutrients in human samples, Faubert et al. confirmed the role of lactate as a TCA carbon source in NSCLC. More recent data reveal that lactate is used as a fuel by diverse cells in the microenvironment under complete aerobic conditions [66]. This use of lactate by tumor cells advantageously reserves glucose for biochemical functions such as the generation of ribose and NADPH by the pentose phosphate pathway that other substrates cannot achieve.

Not only malignant cells convert lactate into glucose. Hepatocytes are involved in converting lactate to glucose and releasing glucose into the bloodstream, where it can be delivered to cells in need of fuel substrates. In addition, lactate can also act as an inter-organ shuttle that feeds the aerobic metabolism and gluconeogenesis pathway. In conclusion, further investigations are needed to determine the source of lactate in circulation. High levels of glucogenic amino acids in the plasma of lung cancer patients, such as tyrosine, threonine, glutamine, valine, and aspartate, reflect the human body’s catabolic state. Indeed, rhabdomyolysis can contribute to maintaining a higher amino acid pool in plasma as depletion of muscle protein mass is a commonly observed problem in patients with cancer [67]. However, caution is needed as no consistency was obtained on the amino acid profiles in lung cancer between different research groups [8,10,68,69,70]. Various factors such as stage of the disease, nutritional status, pathological types, and genotypes may contribute to these conflicting results. The catabolic state of cancer might be confirmed by increased production of the ketoacid β-hydroxybutyrate and glycerol resulting from degradation of peripheral adipose tissue in cancer patients. Glycerol is considered a significant link between fatty acid and carbohydrate metabolism, and the molecule can participate in both the gluconeogenesis and synthesis of lipids that are subsequently incorporated into the cancer cell membranes. The decreased phospholipid plasma levels, which agree with an enhanced membrane synthesis in malignant cells, underscores the role of lipolysis of peripheral fat [35,36]. Glycolysis in lung cancer patients results in higher production of fructose-6-phosphate, a metabolite that can branch glycolysis to enter the hexosamine biosynthesis pathway (HBP) and results in the synthesis of glycoproteins that play a role in regulating growth, differentiation, and metastasis [23].

As previously mentioned, the accurate identification and quantification of plasma metabolites can be challenging in crowded regions of the NMR spectrum. The additional value of higher magnetic field strength (900 MHz versus 400 MHz) was further explored by Louis et al. Through the improved resolution, some overlapping regions in a 400 MHz spectrum can be divided into multiple regions representing a single metabolite [44]. Significantly, these single metabolite signals contribute to more accurate identification of the discriminating metabolites and the elucidation of the underlying disturbed metabolic pathways. As previously described, the plasma concentration of glutamine, glucose, glycerol, (iso)leucine, N-acetylated glycoproteins, threonine, and valine increases in lung cancer patients’ plasma samples, whereas concentrations of alanine, citrate, lactate, non-chlorinated lipids, sphingomyelin, and phosphatidylcholine are decreased. Additionally, the 900 MHz experiments identified and quantified the ketoacid β-hydroxybutyrate, aspartate, and lysine that was impossible using a 400 MHz NMR-spectrometer. Despite the improved spectral resolution, the researchers found a similar discriminative power from the model constructed by data from both NMR spectrometers [44]. 

As previously mentioned, TSP is frequently used as a chemical shift and integration standard. However, TSP binds to human serum albumin which causes fluctuations in the chemical shift and intensity of the TSP signal. Derveaux et al. described the methyl signal of alanine as an optimal standard to calibrate the chemical shift ppm scale, and a known concentration of maleate as an ideal internal standard to quantify the metabolites [49]. The lung cancer versus control classification model obtained by Derveaux et al. resulted in better model diagnostics (specificity of 93%, a sensitivity of 85%, and an area under the curve of 0.95) as compared to the model of Louis et al. (specificity of 92%, a sensitivity of 78%, and an AUC of 0.88) [7,49]. Similarly, the robustness of the classifier was demonstrated in an independent validation cohort. 

In order to evaluate the potential of metabolite biomarkers for diagnosing lung cancer and increasing the effectiveness of clinical interventions, Vanhove et al. investigated the metabolic differences in the blood plasma of patients with suspicious lung abnormalities on CT (computed tomopgraphy) and PET-CT [9]. A LASSO approach was introduced to avoid overfitting the PLS-DA model to select the essential differentiating variables. In conclusion, their results strongly suggested the role of glutamate as a selective inflammatory marker in lung diseases. However, before possible clinical implementation, a more extensive prospective study with external validation is obligatory, and the potential of glutamate as a single biomarker needs to be confirmed by another analytical technique such as HPLC-MS (high performance liquid chromatography-mass spectrometry). A similar research question concerning ground glass opacities (a hazy increase in lung density without obscuration of the underlying vessels or bronchial walls as seen in inflammatory and infectious lung disorders) was investigated in a small study by Li et al. [10]. Similarly, the investigators demonstrated early metabolic alterations differentiating between malignant and benign ground-glass opacities. However, in contrast with Vanhove et al., the metabolic panel of this research group, using mass spectrometry, did not detect glutamate, indicating that further research remains mandatory.

### 3.2. Prognosis and Treatment Responses

Fundamental factors influencing treatment decisions are the expected prognosis, the presence of oncogenic drivers such as, e.g., *EGFR*, *ALK*, *ROS*, and, more recently, the presence of immune checkpoint molecules such as programmed death-ligand-1 [71]. The most established prognostic factor is the tumors–nodes–metastasis (TNM) system [72]. However, the TNM classification does not provide information on the biological behavior of the disease. In addition to the TNM classification, other factors, such as performance status, histology, age, and gender, have been used to predict the aggressiveness of lung cancer [73]. With the advances in molecular and genetic techniques, various features and genetic signatures have been proposed as prognostic factors [74,75,76]. However, none of them have yet been incorporated into clinical practice. Nowadays, researchers focus increasingly on the metabolic alteration of cancer cells, and metabolomics has become a powerful emerging technology to study cancer biology [3,41,42,43]. Nevertheless, the prognostic role of metabolite biomarkers has not been further explored.

The metabolic composition of biofluids or tissue may bring vital insight to stratify patients according to their responses to treatment and ultimately improve therapeutic outcomes. Immune checkpoint inhibitors alone or combined with platinum-based chemotherapy have drastically changed the first-line treatment in locoregional advanced and metastatic lung cancer [77,78]. However, the clinical response to treatment for an individual patient is often unpredictable, and only a small portion of patients benefit from the platinum-based chemotherapeutic component. Therefore, identifying patients who will respond to platinum-based chemotherapy has essential significance and may enable physicians to avoid the risk of toxicity. Over the past decades, much effort has been made to find diagnostic metabolic profiles in distinct cancer types [42,43,54,79]. However, studies regarding the efficacy of systemic treatment through a metabolomic approach are rare. Cisplatin resistance is a major problem in the treatment of lung cancer. Recently, Wangpaichitr et al. discovered that cisplatin-resistant lung cancer cells are no longer addicted to the glycolytic pathway but rely on a higher uptake of glutamine [80]. Activation, expansion, and immune function of immune cells require similar metabolic pathways as cancer cells, with a specific dependence on glycolysis [81,82]. Therefore, competition between immune cells and cancer cells can lead to an inability to clear tumor antigens. However, glycolysis inhibition does not negatively impact the immune function since glucose uptake is higher in glucose-addicted cancer cells [83]. Recently Peng et al. performed a metabolomics approach to discover biomarkers associated with the response to platinum chemotherapy in patients with lung cancer [84]. The results demonstrated that metabolites involved in the TCA cycle, glutamate metabolism, and amino-acid metabolism reflected the response of platinum-containing chemotherapy with a sensitivity and specificity of 100%. However, considering the small number of patients, these findings need confirmation in large patient cohorts. In addition, as immunotherapy has become a new standard of care for lung cancer treatment, a metabolic profile predictive of response is eagerly awaited.

### 3.3. Targeted Treatment of Lung Cancer Based on Metabolic Vulnerabilities

The increased dependence of lung cancer cells on glucose uptake and upregulated glycolysis provides an interesting biochemical basis for developing systemic anticancer treatments that preferentially target cancer cells by pharmacological inhibition of glycolysis. Glucose transporters and glycolytic enzymes are overexpressed in lung cancer, and inhibition of these targets provides an attractive avenue for developing new anticancer treatments [85]. Several inhibitors of glycolytic enzymes and transporters are in preclinical development, yet none of them have approved status [4]. However, the extent of metabolic reprogramming of malignant cells goes far beyond the glycolytic behavior, encompassing nearly all metabolic routes, including glutaminolysis, lipogenesis, gluconeogenesis, the pentose phosphate pathway, and even oxidative phosphorylation [4,30,36]. Furthermore, several factors, such as mutations, oxygenation, and nutrient availability, contribute to the metabolic phenotype in cancer. Therefore, the individual cancer NMR metabolic profile documentation must be investigated to identify individual therapeutic targets. Indeed, most metabolism-altering agents have been tested without knowing the metabolic profile. In addition, tumoral metabolism seems heterogenic, and therefore a combination strategy might be mandatory.

## 4. Conclusions

There is a tremendous need for diagnostic, prognostic, and predictive biomarkers for cancer. Metabolomics has the potential to impact these areas of oncology, and several research groups have described a plasma metabolite signature to identify lung cancer patients. However, the so-far-inconclusive findings underscore that translating metabolomics from bench to bedside remains a major challenge. One of these challenges of metabolomics is the number and chemical complexity of metabolites in plasma. Indeed, the metabolite composition manifests both a pathological and compensatory metabolism that is also impacted by age, gender, and other factors. Aside from a metabolic phenotype of cancer, there is also a need to define the normal metabolic profile as a point of comparison. Despite the availability of the human metabolome database, inconsistent results of metabolomic studies underscore the importance of a robust sampling and measuring protocol. As standardized (pre)analytical and NMR protocols become more widely available, metabolomics may play an essential role in diagnosis and prognosis, and reveal metabolites that may be potential targets for treatment. 

## Figures and Tables

**Figure 1 ijms-23-05602-f001:**
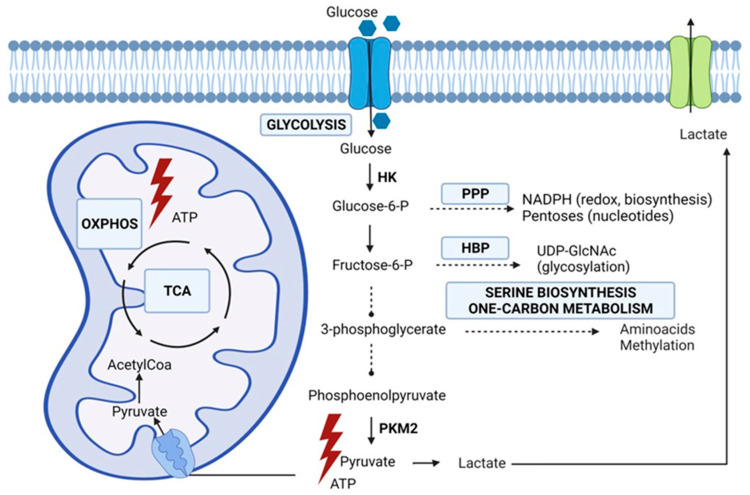
Glucose metabolic pathways emanating from glycolysis involved in lung cancer cell metabolism. HBP (hexosamine biosynthesis), HK (hexokinase), OXPHOS (oxidative phosphorylation), PKM2 (pyruvate kinase isoform M2), PPP (pentose phosphate pathway), TCA (tricarboxylic acid) cycle. Original figure drawn with Biorender.

**Figure 2 ijms-23-05602-f002:**
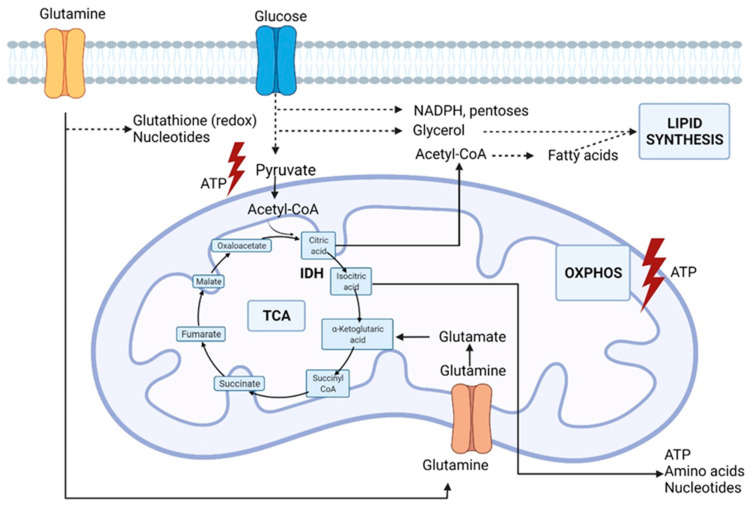
Glutamine metabolic pathways involved in lung cancer cell metabolism. IDH (isocitrate dehydrogenase), OXPHOS (oxidative phosphorylation), TCA (tricarboxylic acid cycle). Original figure drawn with Biorender.

**Figure 3 ijms-23-05602-f003:**
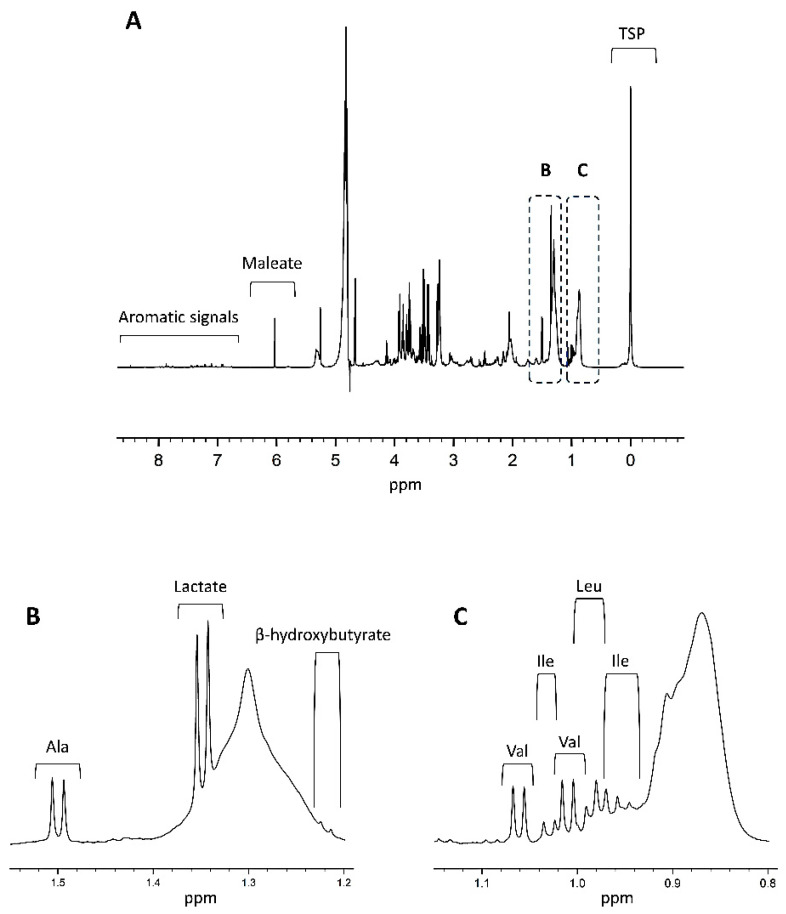
(**A**) Shown is a 600 MHz plasma ^1^H-NMR spectrum of a healthy control, adapted from Ref. [49]. A complete overview of all signals assignments based on spiking experiments can be found in Derveaux et al. [49]. As shown in the figure, the chemical shift values are quoted in ppm and depend on the individual chemical environment of the metabolite’s hydrogen atom. The integrated area under the peak is proportional to the number of hydrogens and thus the metabolite’s concentration. TSP is added as a competitive binder of human serum albumin and ensures dissociation of albumin-bound metabolites. Together with TSP, a fixed concentration of maleate (with a sharp signal around 6 ppm) is added as a reliable internal standard to quantify the human plasma metabolites. Parts (**B**,**C**) zoom in on the 1.2–1.5 ppm and 0.8–1.1 ppm regions of the spectrum, respectively. Clear signals of lactate, alanine, valine, leucine, and isoleucine are highlighted as an example. As shown in (**B**), alanine has a sharp non-overlapping doublet signal (1.5 ppm) that represents its methyl group and efficiently serves as an internal chemical shift reference. Ala: alanine; Ile: isoleucine; Leu: leucine; Val: valine.

**Figure 4 ijms-23-05602-f004:**
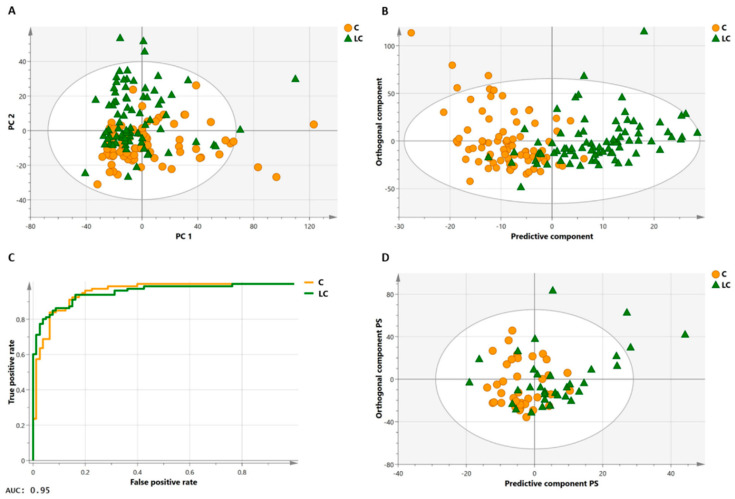
Multivariate statistical analyses on a large lung cancer and control cohort. Principal component analysis (PCA, (**A**)) of the training cohort illustrates the clustering of the two groups (80 controls, orange circles; 80 lung cancer patients, green triangles) in a study performed by Derveaux et al. [49]. In orthogonal partial least squares discriminant analysis (OPLS-DA, (**B**)), a regression model is constructed between the multivariate data and a response variable that contains class information (control, lung cancer). The OPLS-DA score plot in B has a specificity of 93% and sensitivity of 85%, meaning that 85% of all lung cancer patients are correctly classified by the trained OPLS-DA model (trained classifier). The high area under the curve of 0.95 of the receiver operating characteristic (ROC) curve of the trained classifier (**C**) demonstrates that the model has an excellent capability to distinguish between the two groups. An independent subject cohort (**D**) is used to validate the classification model. Here, validation of the OPLS-DA classifier shows a sensitivity and specificity of 74%. Figure adapted from reference [49], open access policy.

## Data Availability

Not applicable.

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
