# Peer review of "Unraveling the Rewired Metabolism in Lung Cancer Using Quantitative NMR Metabolomics"

_ijms, 2022, doi:10.3390/ijms23105602_

Round 1

Reviewer 1 Report

The review described the role of quantitative NMR metabolomics in cancer. The authors may include the tables in which the role of NMR in cancer metabolomics can describes. In the manuscript the figures are adapted as it is from other publication. The authors need to take the permission for same. 

Author Response

Dear reviewer,

Thank you very much for reviewing our manuscript. We greatly appreciate the  constructive comments and suggestions. The sentence concerning the tables is not clear to us : writing the results in a table form or the table of the NMR metabolites with ppm values and patterns ? In case of the latter, we refer to reference 49 as the data are extensive. We added some supplemental figures cfr remark reviewer 2. All the figures are published in accordance with the open policy of the journals and this is now emphasized in the legens of the figures.

Kind regards 

Vanhove Karolien

Reviewer 2 Report

This manuscript is a review of how NMR metabolomics has been used to better understand how metabolism is rewired in lung cancer. The review is informative and worthy of publication. I just have a few minor suggestions for changes and corrections.

on line 59, "principals" should be corrected and replaced by "principles"

Overall this is a valuable manuscript and I would recommend publication following the authors addressing these issues raised in this review

on line 76, "in the net field of ATP..." I presume the authors mean "yield" and not "field", so this should be corrected

when discussing the metabolic pathways impacted in cancer (~ page 4 or so), I think the readers would benefit from a few schematics illustrating how the metabolic pathways are impacted in cancer cells (compared to the metabolism of normal cells). Not everyone has metabolic pathways "at the tip of their fingers" so-to-speak, so illustrations could be quite useful for the general readers.

on line 141, do the authors mean NADPH or NADH? this should be checked to be sure the current wording is correct

line 146, the abbreviation 18-FDG should be defined

The statement starting on linee 235 " one drawback in identifying metabolites in current practice..." is  incorrect and should be reworded. Metabolite assignments and quantitation are done using several NMR parameteers, including chemical shifts, characteristic spectral splitting patterns, and signal intensities - therefore the identification of metabolites is not ambiguous unless there are significant spectral overlaps that can be overcome by spiking of standard compounds and 2D-NMR. The sentence as written is misleading and should be corrected.

the same is true of the sentence starting on line 239 with " To reduce the impact...", NMR is highly reproducible - so if experimental conditions are faithfully replicated, the NMR spectra collected will be hiighly reproducible. It is important to note the difference between experimental/technical variability (which is very small in NMR), and biological variability which is an intrinsic property of different biological samples.

Lastly on lines 358-359, I would suggest that the authors add a reference regarding the nature of the OPLS-DA method 

Author Response

Thank you very much for reviewing our manuscript. We greatly appreciate the constructive comments and suggestions. We have carried out the suggested adaptations and revised the manuscript accordingly. 

We corrected the spelling mistakes in line 59 (principals = principles) and 76 (field = yield) in the original manuscript.

NADPH on line 141 is NADPH, as electrons are transferred from NADPH to NADH by transhydrogenase.

FDG is defined as fluorodeoxyglucose (line 146 in the original text).

We added a book reference to provide more background about the OPLS-DA method : “Eriksson L BT, Johansson E, Trygg J, Vikström C. Multi- and Megavariate Data Analysis Basic Principles and Applications: Umetrics Academy; 2013”.

To illustrate the disturbed glucose metabolism in lung cancer patients we added a figure (Figure 1 in the new document) that was drawn by the first author (copyright first author) and was published in Frontiers of Oncology “Vanhove et al. The Metabolic Landscape of Lung Cancer: New Insights in a Disturbed Glucose Metabolism. Front Oncol. 2019 Nov 15;9:1215. doi: 10.3389/fonc.2019.01215”. The statement starting on line 235 starting with “one drawback in identifying metabolites in current practice..." is replaced by a new text : “ One drawback in identifying metabolites in current practice is the absence of standardized collection and measurement protocols for different biofluids. Varying chemical shift values reported for different matrices or non-human species complicate correct signal assignment (54-56). In addition, these shifts depend on conditions such as pH, concentration, temperature, and ion content, resulting in less accurate identification and quantification if not exactly replicated. One way to correctly determine chemical shifts of the plasma metabolites in the used experimental conditions is by spiking identical plasma from a large plasma pool with different, known metabolites in the same conditions (57)”. The sentence starting on line 239 starting with “to reduce the impact….” was replaced by “Using a standardized protocol, NMR spectroscopy is a very accurate, quantitative, and highly reproducible method with minimal impact of experimental variability on the collected spectra”. Please note that reference 57 and 58 were identical.

Kind regards

Vanhove Karolien
